# UMAMI: Unifying Masked Autoregressive Models and Deterministic Rendering for View Synthesis

**Thanh-Tung Le**[*1]   **Tuan Pham**[*1]   **Tung Nguyen**[2]
**Deying Kong**[3]   **Xiaohui Xie**[1†]   **Stephan Mandt**[1†]
[1]UCI   [2]UCLA   [3]Google

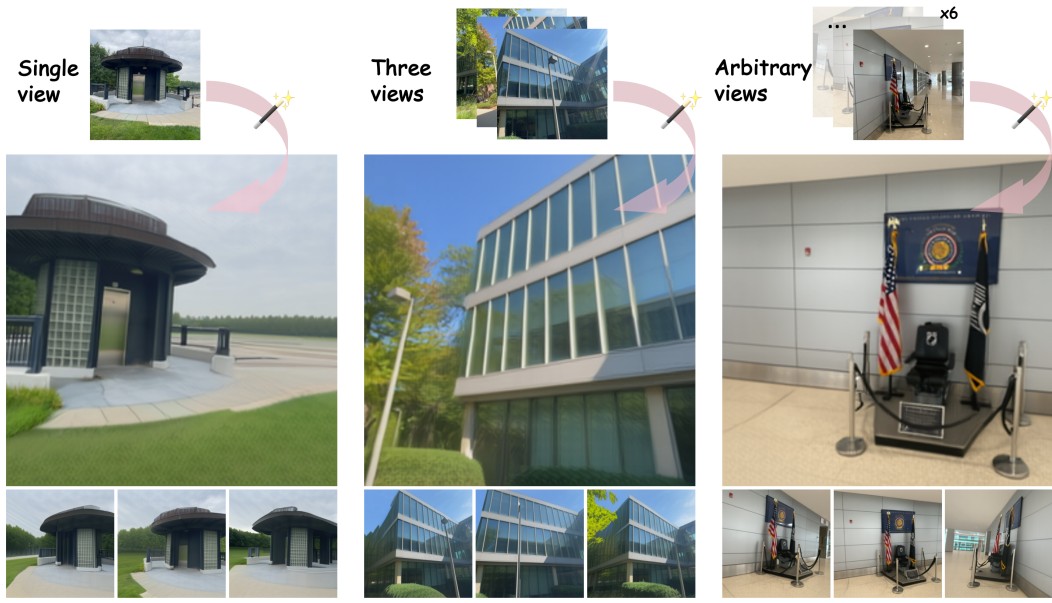

Figure 1: `UMAMI` synthesizes photorealistic novel views from sparse inputs. Shown are single-view generation, three-view extrapolation, and six-view reconstruction. The hybrid model fuses deterministic rendering with diffusion-based completion for unseen regions, yielding fast and consistent results without explicit 3D priors.

## Abstract

Novel view synthesis (NVS) seeks to render photorealistic, 3D-consistent images of a scene from unseen camera poses given only a sparse set of posed views. Existing deterministic networks render observed regions quickly but blur unobserved areas, whereas stochastic diffusion-based methods hallucinate plausible content yet incur heavy training- and inference-time costs. In this paper, we propose a hybrid framework that unifies the strengths of both paradigms. A bidirectional transformer encodes multi-view image tokens and Plücker-ray embeddings, producing a shared latent representation. Two lightweight heads then act on this representation: (i) a feed-forward regression head that renders pixels where geometry is well constrained, and (ii) a masked autoregressive diffusion head that completes occluded or unseen regions. The entire model is trained end-to-end with joint photometric and diffusion losses, without handcrafted 3D inductive biases, enabling scalability across diverse scenes. Experiments demonstrate that our method attains state-of-the-art image quality while reducing rendering time by an order of magnitude compared with fully generative baselines.

* These authors contributed equally to this work. † denotes corresponding authors.

39th Conference on Neural Information Processing Systems (NeurIPS 2025).

# 1 Introduction

Novel view synthesis (NVS) has long been a central problem in computer vision and graphics, aiming to generate realistic, 3D-consistent images of a scene from new camera viewpoints, using a given set of input views with known poses. Traditional methods often require dense input views, treating NVS as a sequential 3D reconstruction and rendering task [49, 34]. Recently, modern deep network priors [98, 11, 21] have been proposed to address the sparse views reconstruction problems, and achieve realistic rendering results.

Two dominant strategies have emerged for sparse-view NVS using deep networks: deterministic and generative-based methods. Deterministic methods often build generalizable networks that predict novel views by incorporating explicit 3D inductive biases [98, 11, 10, 83] or by leveraging priors from large-scale reconstruction models [27, 38, 31] with minimum inductive bias. While these approaches can be effective and fast in rendering observed regions, they often struggle with uncertainty in unobserved areas, leading to blurry predictions. Conversely, generative NVS approaches [87, 102, 21] can generate plausible content for unseen regions. These methods typically employ pretrained diffusion models conditioned on input views and camera poses. However, despite their strong generative capabilities, they often require extensive training data and computational resources, and their iterative sampling process leads to slow rendering speeds.

In this work, we address the question: *"Can we combine the rendering efficiency of deterministic models with the generation capabilities of generative models?"* We aim to unify these disparate approaches into a single, efficient framework. We observe that conventional diffusion models [102, 21], which iteratively generate full images using large UNet or Transformer backbones, can be inefficient if significant portions of the target view are already observable and could be rendered directly by a feed-forward network.

To this end, we introduce UMAMI, **U**nifying **M**asked **A**utoregressive **M**odels and Determin**I**stic Rendering for View Synthesis, a novel hybrid framework for NVS from sparse inputs. Our approach integrates a masked autoregressive model, trained with a diffusion loss [40], alongside a deterministic rendering head. Specifically, drawing inspiration from recent feed-forward NVS models [31], we employ a transformer with bidirectional attention. The model encodes input multi-view images tokens and masked target image tokens, conditioned on Plücker ray embeddings for both input and target views to a represenation. The learned representation fulfills a dual role: (1) it conditions a lightweight MLP diffusion backbone that reconstructs unobserved regions through a diffusion loss [40], and (2) it serves as input to another MLP that directly renders pixel intensities for observed regions, trained with a photometric loss. Our method is designed to be purely data-driven, minimizing reliance on predefined inductive biases in its representation and rendering. This "inductive bias-free" design promotes scalability and generalizability, advantages empirically supported by prior work [21, 102, 31]. Ultimately, UMAMI aims to achieve accurate, training-efficient, and scalable novel view synthesis with photorealistic quality, enjoying both rapid rendering and robust generative completion.

We comprehensively evaluate our model through extensive experiments on RealEstate10K [103] and DL3DV [41], demonstrating competitive performance across both interpolation and extrapolation settings, and under varying input-view configurations.

Our contributions as as follows:

- A hybrid framework for NVS: We propose UMAMI, a novel hybrid architecture that combines deterministic and diffusion-based generation to effectively synthesize both visible and occluded regions from sparse views.

- We demonstrate that UMAMI achieves state-of-the-art performance across multiple benchmarks and input settings, while offering favorable trade-offs between speed and quality.

# 2 Related works

Novel view synthesis (NVS) is a rapidly advancing field. This section summarizes key prior works most relevant to our approach, with a more exhaustive review provided in the Appendix.

## 2.1 Novel view synthesis (NVS)

Novel view synthesis (NVS) has traditionally relied on image-based rendering that blends reference views with proxy geometry [14, 25, 67], light-field techniques that sample the plenoptic function from dense inputs [13], and learning-based variants that predict blending weights or depth maps with CNNs [12]. While multiview-stereo reconstructions enlarge the valid viewing volume [28, 7, 53], the breakthrough NeRF model introduced a differentiable volumetric representation whose photometric training signal became the new benchmark for NVS [49]. Subsequent work has pushed NeRF toward higher fidelity [1, 78, 2], faster inference [57, 24, 58], and fewer input views [50, 84], or has hybridized it with explicit structures such as dense or sparse voxels [72, 42, 20], low-rank decompositions and hashing [4, 8, 9], or point/gaussian primitives [92, 100, 19, 34]. Despite significant progress in rendering quality, these per-scene optimization methods often suffer from slow training times and limited generalization to novel scenes.

## 2.2 Deterministic NVS

To address the limitations of per-scene optimization, deterministic NVS methods train a single network across multiple scenes for fast, feed-forward inference. Some approaches, such as PixelSplat [6], MVSplat [11], and NoPoSplat [96], learn to predict 3D Gaussian parameters directly. While efficient, their reliance on specific 3D representations (e.g., NeRF [49], 3D Gaussians [34]) can hinder scalability. Alternatively, data-driven methods like LVSM [31] and SRT [63] leverage Transformer-only backbones to map input images and target poses to novel views, demonstrating the potential to synthesize views without explicit 3D representations given sufficient data and careful network design. Although scalable and fast, the deterministic nature of these methods typically restricts view generation to regions observed in the input context. Our method, in contrast, aims to synthesize novel views even when parts of the scene are occluded or outside the context views.

## 2.3 Generative NVS

In addition to deterministic approaches, generative approaches have adapted powerful image and video diffusion models (DMs) [3, 69] for NVS [102, 80, 87, 36], leveraging their strong generative priors. Early diffusion-based NVS models [66, 44, 48] often utilized image DMs conditioned on input images. Contemporary methods increasingly adopt video DMs [21, 102], conditioned on camera poses, to achieve finer-grained control and generate high-quality views of unseen regions. However, training these large-scale generative models demands substantial data and computational resources, potentially impacting rendering performance.

## 3 Background

### 3.1 Novel View Synthesis

**Deterministic approaches** focus on learning a mapping $f_\theta(\mathbf{I}^{\text{ctx}}, \boldsymbol{\pi}^{\text{ctx}}, \boldsymbol{\pi}^{\text{tgt}})$ that directly generates the target image $\mathbf{I}^{\text{tgt}}$. Here, $\mathbf{I}^{\text{ctx}}$ and $\mathbf{I}^{\text{tgt}}$ represent context and target images, while $\boldsymbol{\pi}^{\text{ctx}}$ and $\boldsymbol{\pi}^{\text{tgt}}$ denote their respective camera poses. This mapping $f_\theta$ may be realized through pure neural networks [31, 71] or by integrating 3D inductive biases [98, 6, 11, 96]. Although generally efficient, a fundamental limitation of deterministic methods is the inability to generate unseen region due to the deterministic nature.

**Generative approaches** learn to sample $\mathbf{I}^{\text{tgt}}$ from a learned conditional distribution $p_\theta(\mathbf{I}^{\text{tgt}}|\mathbf{I}^{\text{ctx}}, \boldsymbol{\pi}^{\text{ctx}}, \boldsymbol{\pi}^{\text{tgt}})$. This distribution is often modeled using powerful generative frameworks such as diffusion models [102, 21]. The advantage of such generative techniques lies in their ability to convincingly hallucinate regions absent in the input views. Nevertheless, this capability comes at a significant computational cost for both training and inference, thereby posing challenges to their widespread practical use in NVS applications.

### 3.2 Masked Autoregressive Image Generatation

Unlike diffusion models, autoregressive (AR) models [76, 51, 22] approach the generation of an ordered token sequence $\{x^1, x^2, \ldots, x^n\}$ (with $1 \leq i \leq n$ defining the order) by formulating

the problem as "next token prediction." This is mathematically expressed by factorizing the joint probability:

$$p(x^1, \ldots, x^n) = \prod_{i=1}^{n} p(x^i | x^1, \ldots, x^{i-1}), \tag{1}$$

where the conditional probability $p(x^i | x^1, \ldots, x^{i-1})$ is modeled by a neural network.

Departing from traditional AR methods [22, 51], the Masked Autoregressive (MAR) model [40] presents an different approach that unifies random-order AR principles with masked generative modeling through the use of a Diffusion Loss. In MAR, an autoregressive network produces a feature vector $z = f(\cdot) \in \mathbb{R}^D$. This vector, alongside a small MLP $\epsilon_\theta(\cdot)$, is used to model the conditional distribution $p(x|z)$ for a token $x \in \mathbb{R}^d$. The model is trained using the denoising criterion:

$$\mathcal{L}(z, x) = \mathbb{E}_{\epsilon, t} \left[ ||\epsilon - \epsilon_\theta(x_t | t, z)||^2 \right], \tag{2}$$

where $\epsilon \in \mathbb{R}^d$ is Gaussian noise and $t \in \mathbb{R}$ is the timestep.

Compared to traditional diffusion models [59, 52], MAR sample an image by iteratively unmasking tokens using the MLP diffusion conditioned on learned latent from transformer. MAR demonstrates computational efficiency while showcasing competitive performance with its counterparts.

Building upon MAR's efficient generative capabilities, our work introduces a novel hybrid method for the NVS task. Specifically, we leverage MAR's generative framework within a hybrid network that incorporates deterministic rendering. Furthermore, we propose a unique sampler specifically designed to efficiently generate novel views, thereby avoiding the iterative full-image generation typical of large backbone architectures [102, 21]. This approach enables our generative solution to achieve rendering speeds an order of magnitude faster than previous generative NVS methods. We believe this to be the first proposal of a hybrid method that successfully unifies a generative model with a deterministic head to tackle the NVS challenge.

# 4 Methods

In this section, we first outline our problem formulation (Section 4.1) and then details our hybrid model (Section 4.2). Subsequently, we present the training loss (Section 4.3) and conclude by proposing a novel hybrid sampler (Section 4.4).

## 4.1 Problem Formulation

Given sparse input images with known camera poses $\{(\mathbf{I}^{\text{ctx}}, \boldsymbol{\pi}^{\text{ctx}})\}$, our goal is to model the conditional distribution $p(\mathbf{I}^{\text{tgt}} | \mathbf{I}^{\text{ctx}}, \boldsymbol{\pi}^{\text{ctx}}, \boldsymbol{\pi}^{\text{tgt}})$ to synthesize realistic novel view $\mathbf{I}^{\text{tgt}}$ given its camera poses $\boldsymbol{\pi}^{\text{tgt}}$.

**Data Representation** To jointly encode image content and camera pose information, we follow the established convention of concatenating each image with its corresponding Plücker ray embeddings [54] along the channel dimension. This concatenated representation is subsequently processed through an MLP-based tokenizer to produce discrete latent tokens. For simplicity, we forego a VAE-based approach and directly tokenize each image into $8 \times 8$ patches. We denote the resulting token sequence from context images and their poses $\{(\mathbf{I}^{\text{ctx}}, \boldsymbol{\pi}^{\text{ctx}})\}$ as $\mathbf{c} = (c^1, c^2, ..., c^N)$; and similarly define the target token sequence from $\{(\mathbf{I}^{\text{tgt}}, \boldsymbol{\pi}^{\text{tgt}})\}$ as $\mathbf{x} = (x^1, x^2, ..., x^M)$. Ignoring the facts that the target camera pose are also embedded within $\mathbf{x}$ and treat them purely as image tokens, we can write the target conditional distribution as $p(\mathbf{x}|\mathbf{c})$.

## 4.2 Hybrid Masked Autoregressive Models for Novel View Synthesis

As discussed in Section 3, deterministic-based NVS approaches [31, 63] model $p(\mathbf{x}|\mathbf{c})$ to be a deterministic function $F$ of inputs: $p(\mathbf{x}|\mathbf{c}) = \delta(\mathbf{x} - F(\mathbf{c}))$, where $\delta$ is the Dirac delta function. While they have shown strong performance in generating high-fidelity outputs for regions covered by input views, they struggle to handle unseen regions due to their inability to model inherent ambiguity. In contrast, generative models based on diffusion [21, 102, 80] can generate plausible completions for unobserved regions, but often incur significantly higher computational costs due to iterative sampling over the full image. This trade-off motivates our hybrid design in UMAMI, which is based on the factorization:

$$p(\mathbf{x}|\mathbf{c}) = \delta(\mathbf{x}_D - F(\mathbf{c})) \cdot p(\mathbf{x}_S | \mathbf{x}_D, \mathbf{c}) \tag{3}$$

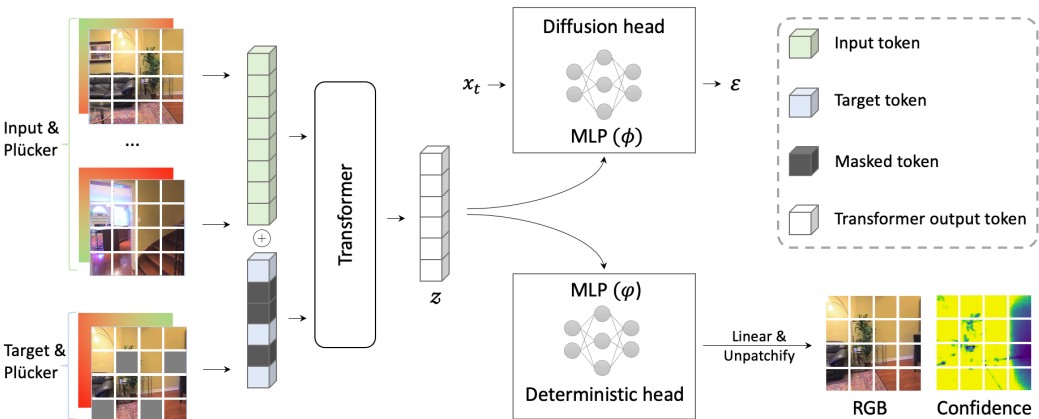

Figure 2: UMAMI **synthesizes target images from their camera poses and context views (each paired with its Plücker pose).** During training, we randomly mask the target image, replace masked areas with learnable tokens, and concatenate these with the target's Plücker embedding. Input views are also tokenized. A Transformer processes both tokenized inputs and the masked target representation to produce a latent $z$. This code inputs to two MLP heads: a deterministic head ($\varphi$) outputs RGB and confidence, while a diffusion head ($\phi$) models the distribution of target tokens conditioned on $\mathbf{z}$. The model is trained end-to-end using a weighted loss combination (Section 4.3). At inference, the target image is initialized with learned masked tokens for our proposed hybrid sampling (Section 4.4).

where $\mathbf{x}_D$ and $\mathbf{x}_S$ are disjoint subsets of $\mathbf{x}$ such that $\mathbf{x} = \mathbf{x}_D \cup \mathbf{x}_S$. Intuitively, $\mathbf{x}_D$ corresponds to the tokens that are fully determined by the input context $\mathbf{c}$ (e.g., seen or deterministically visible regions) and can be computed directly as a function $F(\mathbf{c})$. In contrast, $\mathbf{x}_S$ represents tokens in uncertain or unseen regions, which require sampling from a complex conditional distribution $p(\mathbf{x}_S | \mathbf{x}_D, \mathbf{c})$.

**Model Architecture** The architecture of UMAMI is illustrated in the Figure 2. UMAMI is a masked autoregressive model designed to support both efficient deterministic prediction and flexible stochastic generation by progressively unmasking target tokens. At the core of our model is a transformer backbone [77] that extract the target latent representation $\mathbf{z}$ from the partially masked $\mathbf{x}$ and context $\mathbf{c}$. Following previous works [21, 31], we adopt a decoder-only, bi-directional transformer backbone.

To generate the target tokens, UMAMI uses two specialized output heads. The **deterministic head**, parameterized by $\varphi$ in Figure 2, calculates $F(\mathbf{x})$ using the extracted latents $\mathbf{z}$ from the transformer backbone and reconstructs tokens in $\mathbf{x}_D$ in a single forward pass, leveraging regions of high confidence inferred from the context. In contrast, the **diffusion head**, parameterized by $\phi$ in Figure 2, models the conditional distribution over $\mathbf{x}_S$ and performs iterative denoising to progressively generate plausible content in uncertain or unseen regions. Following MAR [40], both heads are small MLP networks with SiLU activation [18] that operate on each token latent individually, and the diffusion head takes an additional time embedding as input. This dual-head design enables UMAMI to adaptively combine the speed and accuracy of deterministic prediction with the generative capacity of diffusion models, effectively addressing both observed and novel view synthesis scenarios.

In practice, the separation between deterministic and uncertain regions is not known a priori. To address this, we introduce a pixel-wise confidence prediction that estimates an uncertainty score for each pixel. The confidence score of a patch is defined as the minimum confidence among its pixels. Given a threshold $\tau$, patches with confidence above that threshold are assigned to $\mathbf{x}_D$, while the remaining are treated as $\mathbf{x}_S$ and handled via the stochastic generation process.

## 4.3 Training Losses

We train UMAMI using a masked autoregressive generative framework [40, 5, 39]. At each training step, a binary mask $\mathbf{m}$ is sampled uniformly to mask a subset of the target image patches. Crucially, only the target image is masked (e.g. each selected patch is replaced with a learnable token) while the corresponding target camera pose embeddings are preserved. The model is then optimized to reconstruct the masked patches conditioned on the context and unmasked target information, using a combination of deterministic and diffusion losses:

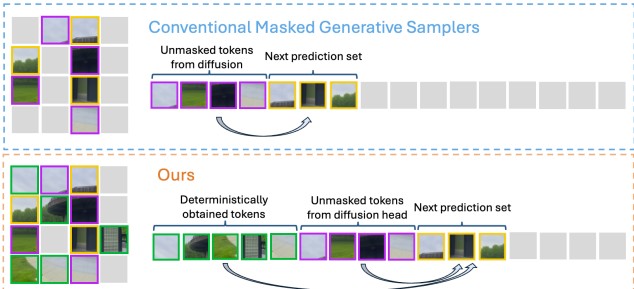

Figure 3: **Hybrid Masked Autoregressive Sampler.** Top: Conventional Masked Generative samplers [40, 5, 39] predict multiple tokens simultaneously using random ordering. Bottom (Ours): A deterministic first pass for high confidence tokens, followed by simultaneous random-order sampling for the remaining tokens, significantly boosts rendering times for the NVS task.

**Deterministic reconstruction loss**    The deterministic head produces token predictions $\hat{\mathbf{x}}$ for masked patches, which yields reconstruction of the target images $\hat{\mathbf{I}}^{\text{tgt}}$. To supervise this process, we employ standard photometric losses for novel view synthesis, defined as:

$$\mathcal{L}_{\text{render}} = \text{MSE}(\hat{\mathbf{I}}^{\text{tgt}}, \mathbf{I}^{\text{tgt}}) + \lambda_{\text{p}}\text{Perceptual}(\hat{\mathbf{I}}^{\text{tgt}}, \mathbf{I}^{\text{tgt}}), \tag{4}$$

where $\lambda_p$ is the weight for balancing the perceptual loss [33]. Importantly, this loss is computed over the full image rather than individual patches to encourage spatial consistency.

**Confidence loss**    As discussed in Section 4.2, we augment the deterministic head to output a pixel-level confidence map $\mathbf{s}_p$, where each value indicates the model's confidence in its prediction. The confidence-aware loss balances the regression error with a regularization term that penalizes overconfidence:

$$\mathcal{L}_{\text{conf}} = \mathbf{m} \odot (\mathbf{s}_p \odot ||\hat{\mathbf{I}}^{\text{tgt}} - \mathbf{I}^{\text{tgt}}||_2^2 - \lambda_s \cdot \log \mathbf{s}_p), \tag{5}$$

where $\lambda_s$ is a hyper-parameter controlling the regularization term [81]. The loss is averaged over all masked parts. We compute a patch-wise confidence map $\mathbf{s}$ by taking the minimum value of $\mathbf{s}_p$ within each patch.

**Diffusion loss**    To model the conditional distribution over uncertain tokens, we incorporate a diffusion model following the formulation of DDPM [26]. Specifically, we use a linear noise schedule to corrupt the ground truth tokens and train the model to reverse this process. Given a noisy token $x_t$ at timestep $t$ and its corresponding latent $z$ extracted from the transformer backbone, the diffusion head predicts the added noise $\hat{\epsilon}$. The denoising objective is defined over all masked tokens:

$$\mathcal{L}_{\text{diff}} = \mathbb{E}_{\epsilon,t} \left[ ||\epsilon - \hat{\epsilon}(x_t|t, z)||_2^2 \right], \tag{6}$$

where $\epsilon \sim \mathcal{N}(0, I)$ is Gaussian noise, and $t$ is sampled uniformly over the diffusion steps.

To better allocate learning effort, we emphasize uncertain regions during training by predicting a token-wise weighting scheme derived from the patch-level confidence map $\mathbf{s}$. Specifically, we define the weight for each token as $\max(\mathbf{s}, \lambda_d)/\lambda_d$, where $\lambda_d$ is a hyperparameter. This weighting encourages the model to focus more heavily on regions with lower confidence, enhancing generative quality in areas with higher ambiguity.

**Total loss**    Our model is trained end-to-end using a weighted sum of the aforementioned losses.

### 4.4 Hybrid Masked Autoregressive Sampling

The overall sampling process is illustrated in Figure 3. Given a masked target image and its corresponding camera pose, `UMAMI` performs hybrid inference by first identifying and reconstructing the set of deterministic tokens $\mathbf{x}_D$, and then generating the remaining uncertain tokens $\mathbf{x}_S$ through a diffusion-based process.

In the first stage, the model performs a single forward pass through the deterministic head to predict $\mathbf{x}_D$, guided by the confidence map predicted from the transformer backbone. Tokens with confidence scores greater then a predefined threshold $\tau$ are reconstructed deterministically.

In the second stage, the remaining masked tokens $\mathbf{x}_S$, are iteratively sampled using the diffusion head. We employ a cosine unmasking schedule following the approach of [40], which gradually reveals

Table 1: Quantitative results on RealEstate10K across different validation splits. Best results are highlighted in red, second-best in orange.

| Method | Params (M) | Re10K-2view-extra | | | Re10K-2view-interp | | | Re10K-3view | | |
|---|---|---|---|---|---|---|---|---|---|---|
| | | PSNR ↑ | LPIPS ↓ | SSIM ↑ | PSNR ↑ | LPIPS ↓ | SSIM ↑ | PSNR ↑ | LPIPS ↓ | SSIM ↑ |
| **Deterministic** | | | | | | | | | | |
| MVSplat [11] | 12.0 | 23.30 | 0.160 | 0.830 | 26.39 | 0.128 | 0.869 | 25.64 | 0.142 | 0.857 |
| DepthSplat [91] | 360 | 24.57 | 0.158 | 0.848 | 27.44 | 0.119 | 0.887 | 22.54 | 0.177 | 0.824 |
| LVSM [31] | 171 | 28.51 | 0.117 | 0.882 | 29.67 | 0.098 | 0.906 | 30.04 | 0.090 | 0.936 |
| **Diffusion-based** | | | | | | | | | | |
| ViewCrafter [99] | N/A | - | - | - | 21.42 | 0.203 | 0.710 | 22.81 | 0.164 | 0.830 |
| SEVA [102] | 1300 | 24.00 | 0.100 | 0.797 | 25.66 | 0.061 | 0.847 | 27.57 | 0.073 | 0.892 |
| UMAMI | 271 | 28.95 | 0.107 | 0.897 | 28.85 | 0.101 | 0.899 | 31.06 | 0.084 | 0.946 |

more tokens at later iterations using $T_S$ unmasking steps. As $|\mathbf{x}_S| \leq |\mathbf{x}|$, especially in scenarios where target views significantly overlap with context views, we introduce a dynamic strategy to adjust the number of unmasking steps accordingly. Specifically, given a maximum step budget $T_{\max}$ for unmasking the entire token set $\mathbf{x}$, the number of steps allocated for $\mathbf{x}_S$ is computed using a simple linear scaling rule: $T_S = \lceil |\mathbf{x}_S|/|\mathbf{x}| \cdot T_{\max} \rceil$. The hyperparameter $T_{\max}$ is fixed across experiments, and $T_S$ is automatically determined by the number of tokens to be unmasked. Despite its simplicity, we find this strategy to be effective in practice and well-suited for varying levels of token uncertainty.

## 5 Experiments

**Datasets** We evaluate UMAMI on two scene-level novel view synthesis benchmarks: RealEstate10K (CC-BY-4.0) [103] and DL3DV (CC-BY-4.0) [41]. RealEstate10K consists of 80K indoor and outdoor video clips sourced from YouTube, while DL3DV features over 10K videos captured across a wide range of real-world locations. We train separate models for each dataset at a resolution of $256 \times 256$. For the RealEstate10K dataset, we adopt the evaluation split from PixelSplat [6], which primarily features target views located between the 2 input views, making it suitable for assessing interpolation performance. We refer to this split as *Re10K-2View-Interp*. To evaluate extrapolation ability, we construct a complementary split by swapping the roles of the context and target views, which we denote as *Re10K-2View-Extra*. Additionally, we incorporate the 3-view validation split introduced in Reconfusion [89], labeled as *Re10K-3View*, respectively. For the DL3DV dataset, we follow the validation setup from Zhou et al. [102], using the 1-view, 3-view, and 6-view input configurations, which we name *DL3DV-1View*, *DL3DV-3View*, and *DL3DV-6View*, respectively.

**Experiment Details** Each model is trained for 100K iterations with a batch size of 32, using the AdamW optimizer [47] with a learning rate of $2 \times 10^{-4}$ and a cosine decay schedule. Training takes approximately two days on $8\times$ NVIDIA A100 GPUs. During training, we randomly sample 1 or 2 context views and select between 1 and 3 target views per training example. In the main experiments, we report results for predicting a single target view, while results for generating multiple target views are included in the Appendix. To accelerate convergence, we initialize our model using the pretrained transformer backbone from LVSM [31]. We use a fixed threshold value of $\tau = 0.95$ and a maximum sampling steps $T_{\max} = 32$ across our experiments, as we found those values balance well between generation quality and speed. For diffusion sampling, we use 50 DDPM steps with a CFG value of 2.0 and a sampling temperature of 0.9. An details on other hyperparameters of our model are in the Appendix.

**Baselines** To the best of our knowledge, we are the first method that perform a hybrid render on deterministic and generative method, thus we have no direct competitors. Therefore, we compare our method to different deterministic and generative baselines. For deterministic methods, we compare UMAMI to MVSplat [11], LVSM [31]. For generative approaches, we compare UMAMI with ViewCrafter [99] and SEVA [102].

### 5.1 Experiment results

**Quantitative results** Tables 1 and 2 present a comprehensive comparison of our method UMAMI against both deterministic and diffusion-based baselines on the RealEstate10K and DL3DV datasets, respectively. On RealEstate10K, UMAMI consistently achieves top-tier performance across all splits. On the Re10K-2view-interp split, it matches LVSM closely, trailing by only 0.82 PSNR (28.85 vs.

Table 2: Quantitative results on DL3DV across 1-view, 3-view, and 6-view settings. Best results are highlighted in red, second-best in orange.

| Method | DL3DV-1view | | | DL3DV-3view | | | DL3DV-6view | | |
|---|---|---|---|---|---|---|---|---|---|
| | PSNR ↑ | LPIPS ↓ | SSIM ↑ | PSNR ↑ | LPIPS ↓ | SSIM ↑ | PSNR ↑ | LPIPS ↓ | SSIM ↑ |
| **Deterministic** | | | | | | | | | |
| DepthSplat [91] | 9.63 | 0.580 | 0.349 | 12.52 | 0.405 | 0.452 | 15.72 | 0.481 | 0.513 |
| **Diffusion-based** | | | | | | | | | |
| ViewCrafter [99] | 8.97 | 0.616 | 0.323 | 11.50 | 0.576 | 0.400 | 13.78 | 0.558 | 0.469 |
| SEVA [102] | 13.01 | 0.484 | 0.360 | 15.95 | 0.316 | 0.480 | 17.98 | 0.232 | 0.546 |
| UMAMI | 12.81 | 0.574 | 0.269 | 16.37 | 0.386 | 0.444 | 17.33 | 0.326 | 0.476 |

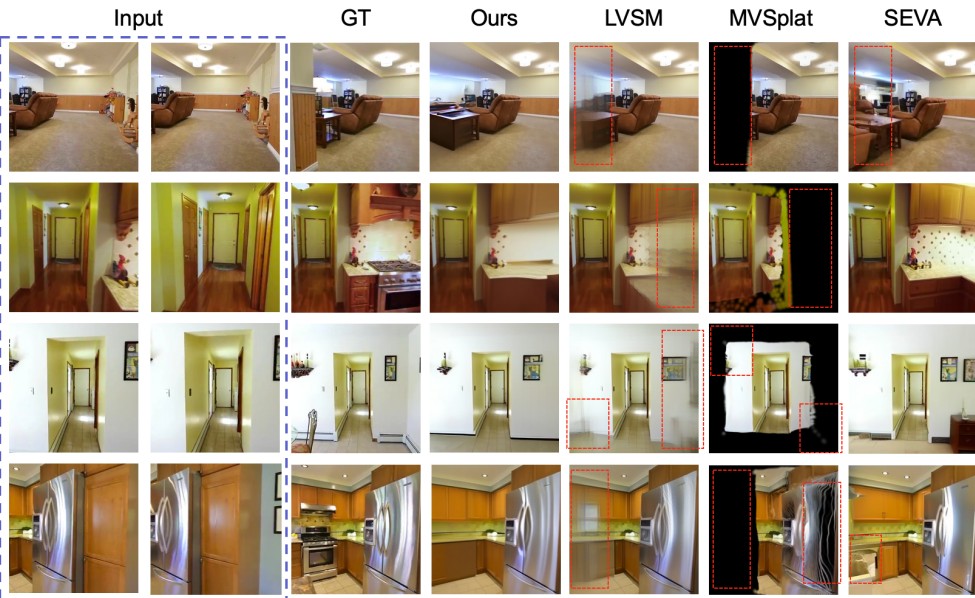

Figure 4: **Qualitative results on Re10K** Evaluation of UMAMI on the challenging *Re10K-2View-Extra* extrapolation set, comparing it with LVSM [31], MVSplat [11], and SEVA [102]. UMAMI not only renders sharp details in observed regions but also generates plausible content for unseen areas. More results can be viewed in the Appendix.

29.67). On the Re10K-2view-extra split, it obtains the highest PSNR (28.95) and SSIM (0.897), outperforming the second-best LVSM by 0.44 PSNR and 0.015 SSIM, while maintaining a second-best LPIPS of 0.107; showing its capabilities of doing extrapolation. On the Re10K-3view setting, UMAMI surpasses all baselines with the best PSNR (31.06) and SSIM (0.946), while having the second best LPIPS of 0.084.

On the DL3DV benchmark, UMAMI delivers competitive performance across all input configurations. It achieves the second-best PSNR in both the 1-view (12.81) and 6-view (17.33) settings, and ranks second in LPIPS across all three input setups. Notably, in the 3-view setting, UMAMI achieves the highest PSNR among all methods. While its SSIM lags behind DepthSplat, UMAMI outperforms ViewCrafter in both the 3-view and 6-view scenarios. These results underscore the robustness and adaptability of our hybrid framework, demonstrating its effectiveness in handling diverse view configurations and maintaining a strong balance between reconstruction fidelity and perceptual quality.

**Varying overlap ratios** Following NoPoSplat [96], we evaluate UMAMI on Re10k test set with varying camera overlaps based on ratio of image overlap: small (0.05%-0.3%), medium (0.3% - 0.55%), and large (0.55% - 0.8%), determined using dense feature matching method, RoMA [17]. The results, shown in the Table 3, demonstrate that UMAMI outperforms NoPoSplat on all metrics and validation sets. As with NoPoSplat, we also observed that performance improves as the overlap ratio increases, which suggests less scene occlusion. Notably, our method remains robust even with a small camera overlap, outperforming NoPoSplat across all metrics. This confirms UMAMI's effectiveness across various datasets and overlap ratios.

Table 3: Quantitative comparison across overlaping ratios (Small, Medium, Large). Best results are highlighted in red, second-best in orange.

| Method | Small | | | Medium | | | Large | | |
|---|---|---|---|---|---|---|---|---|---|
| | PSNR ↑ | SSIM ↑ | LPIPS ↓ | PSNR ↑ | SSIM ↑ | LPIPS ↓ | PSNR ↑ | SSIM ↑ | LPIPS ↓ |
| NoPoSplat | 22.514 | 0.784 | 0.210 | 24.899 | 0.839 | 0.160 | 27.411 | 0.883 | 0.119 |
| Ours | 23.558 | 0.806 | 0.176 | 26.713 | 0.862 | 0.130 | 29.782 | 0.907 | 0.094 |

**Qualitative results** presented in Figure 4, highlight the performance of our method against several methods. Firstly, MVSplat [11], as a deterministic method employing 3D Gaussians, is unable to generate content beyond the provided context images, resulting in black rendered areas in unobserved regions. Similarly, while LVSM [31] avoids such black areas by forgoing 3D inductive biases, its non-generative nature results in blurry predictions for unseen pixels. Our method overcomes these limitations of deterministic approaches, demonstrating the ability to both accurately render observed regions and plausibly generate content in unobserved areas. Finally, in comparison to SEVA [102], a considerably larger model with 1.3B parameters (versus our 271M parameters), our approach achieves comparable performance on visible regions and produces results with fewer artifacts.

## 5.2 Ablation study and Analysis

Table 4: Ablation study on threshold $\tau$. Higher $\tau$ improves image quality but increases transformer calls and runtime.

| $\tau$ | Time (s) | # Trans. Calls | LPIPS ↓ |
|---|---|---|---|
| 0 | 0.02 | 1.00 | 0.398 |
| 0.5 | 2.71 | 12.31 | 0.394 |
| 0.8 | 4.30 | 18.99 | 0.389 |
| 0.9 | 4.62 | 20.30 | 0.387 |
| 0.95 | 4.77 | 21.14 | 0.386 |
| 1 | 7.63 | 33.00 | 0.377 |

Table 5: Ablation study on the number of context views. Increasing the number of context views ($N_c$) improves image synthesis quality by providing more deterministic tokens and reducing the number of average transformer calls, due to higher confidence in a larger portion of the scene.

| $N_c$ | # Deter Tokens | # Trans. Calls | LPIPS ↓ |
|---|---|---|---|
| 1 | 119.24 | 29.55 | 0.574 |
| 3 | 394.68 | 21.14 | 0.386 |
| 6 | 527.25 | 16.98 | 0.326 |

**Effect on threshold $\tau$ during sampling** We control the balance between UMAMI's deterministic and diffusion heads during sampling using a threshold $\tau$ in Table 4. Setting $\tau = 0$ engages only the deterministic head, enabling UMAMI to predict all tokens in $0.02$s with one transformer call. Conversely, setting $\tau = 1$ relies exclusively on the diffusion head for sampling target views, which yields the optimal LPIPS score in our experiments. We observe that incrementally increasing $\tau$ from $0$ to $1$ enhances LPIPS performance, though at the cost of increased runtime due to more frequent transformer and diffusion sampling operations. Thus, by adjusting $\tau$, our dual-head model offers a flexible mechanism to trade off inference speed against generative quality.

**Effect on the number of context views** We conduct an ablation study on the effect of varying the number of context views, as shown in 5. Results indicate that increasing the number of input views leads to a significant boost in mean LPIPS. Furthermore, with more context available, the model exhibits higher confidence, resulting in a greater proportion of tokens being handled deterministically. This, in turn, reduces the number of transformer calls required during sampling, leading to improved computational efficiency.

**Run time analysis** Unlike conventional methods with fixed rendering times, UMAMI offers operational flexibility by adaptively engaging its deterministic and generative heads. For instance, UMAMI renders an image in approximately 5s when $\tau = 0.95$ (details in Table 4). This is considerably faster than generative counterparts like SEVA [102], which takes about 1 minute to sample an image. While purely deterministic methods [11] achieve sub-second rendering, they sacrifice the ability to generate content for unobserved target regions. UMAMI thus provides a compelling trade-off: it achieves strong generative capabilities for a modest increase in runtime compared to deterministic approaches, while remaining significantly more efficient than other generative models.

**Backbone initialization** As mentioned in Section 5, we initialize our model using pretrained weight from LVSM [31]. To further demonstrate the strength of our method, we train the model from scratch

Table 6: Ablation results on DL3DV across 1-view, 3-view, and 6-view settings comparing pretrained and random initialization. Best results are highlighted in red, second-best in orange.

| Method | DL3DV-1view | | | DL3DV-3view | | | DL3DV-6view | | |
|---|---|---|---|---|---|---|---|---|---|
| | PSNR ↑ | LPIPS ↓ | SSIM ↑ | PSNR ↑ | LPIPS ↓ | SSIM ↑ | PSNR ↑ | LPIPS ↓ | SSIM ↑ |
| Pretrained | 12.81 | 0.574 | 0.269 | 16.37 | 0.386 | 0.444 | 17.33 | 0.326 | 0.476 |
| Random Init. | 11.80 | 0.543 | 0.256 | 14.46 | 0.374 | 0.370 | 15.43 | 0.318 | 0.404 |

on DL3DV with randomly initialized weights, without relying on LVSM pretrained on Re10K. This variant shares the same settings as the pretrained version, except for a larger batch size (512 vs. 32) to stabilize training. The results, presented in the Table 6, reveal that even without pretraining, our model performs comparably—showing slightly lower PSNR and SSIM but improved LPIPS. Remarkably, even with random initialization, our method consistently outperforms ViewCrafter and DepthSplat, underscoring that our performance stems from the strength of our hybrid deterministic-generative design, rather than dependence on LVSM initialization and Re10K pretraining.

## 6 Discussion and Conclusion

**Limitations and Future Work**   While our method achieves competitive performance, it also has several limitations. First, because we operate directly in pixel space, each image is represented by a large number of tokens (e.g., $32 \times 32 = 1024$), which increases memory and computational requirements. A promising direction for future work is to adapt our framework to operate in the latent space of a pretrained VAE [35], which would reduce the token count while preserving semantic content. Second, unlike recent diffusion-based NVS approaches [21, 102, 99], our model does not make use of any pretrained text-to-image priors. Integrating such powerful generative priors [40, 16] could enhance the model's ability to hallucinate plausible unseen regions and improve visual fidelity in sparse-view settings. We also leave for future exploration techniques to further accelerate sampling and incorporate temporal consistency for video-based novel view synthesis. On the social impact side, this work could enable deepfake information, so users will be required to follow usage guidelines.

**Conclusion**   We have presented UMAMI, a hybrid framework for novel view synthesis that unifies deterministic and generative modeling to handle both seen and unseen regions effectively. By leveraging a confidence-aware mechanism, our model adaptively allocates computation between a fast deterministic head and a diffusion-based head, achieving a strong balance between efficiency and image quality. Extensive experiments on RealEstate10K and DL3DV demonstrate that UMAMI is competitive with both deterministic and diffusion-only baselines across various input configurations. Our results suggest a promising direction for designing more efficient approaches to novel view synthesis.

## Acknowledgements

Stephan Mandt acknowledges funding from the National Science Foundation (NSF) through an NSF CAREER Award IIS-2047418, IIS-2007719, the NSF LEAP Center, and the Hasso Plattner Research Center at UCI. Xiaohui Xie acknowledges funding from NIH 1P01CA288662-01A1 and Kay Family Foundation. Parts of this research were supported by the Intelligence Advanced Research Projects Activity (IARPA) via the Department of Interior/ Interior Business Center (DOI/IBC) contract number 140D0423C0075. The U.S. Government is authorized to reproduce and distribute reprints for Governmental purposes notwithstanding any copyright annotation thereon. Disclaimer: The views and conclusions contained herein are those of the authors and should not be interpreted as necessarily representing the official policies or endorsements, either expressed or implied, of IARPA, DOI/IBC, or the U.S. Government.

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

# Supplement to "UMAMI: Unifying Masked Autoregressive Models and Deterministic Rendering for View Synthesis"

## A  Failure cases

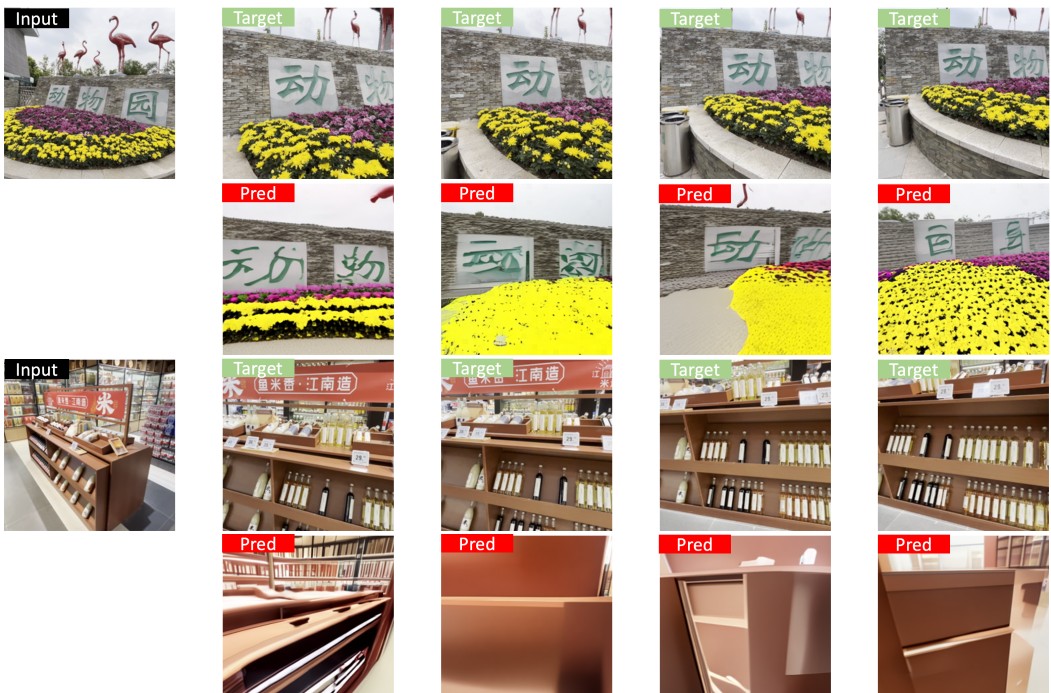

Figure 5: **Failure cases.** Our method may produce noticeable artifacts when target camera poses are too distant from the input view. Increasing the scale of training data and model parameters could improve the robustness of UMAMI.

## B  Related works

### B.1  Feed-forward deterministic NVS methods

Early generalizable methods for Novel View Synthesis (NVS) demonstrated the potential of neural networks, trained across various scenes, to enable fast inference of novel views or underlying 3D representations in a feed-forward manner. Prominent examples include PixelNeRF [98], MVSNeRF [10], and IBRNet [83], which typically predict volumetric 3D representations by incorporating 3D-specific priors like epipolar geometry or plane sweep cost volumes. Subsequent research has extended these capabilities, improving performance particularly under challenging conditions such as sparse input views [45, 32, 29, 30], and adapting these techniques for emerging representations like 3D Gaussian Splatting (3DGS) [6, 73, 11, 74].

Recently, 3D Large Reconstruction Models (LRMs) have emerged [27, 38, 82, 93, 88, 90], leveraging the power of scalable transformer architectures [77] trained on extensive datasets to learn generic 3D priors. While these methods successfully avoid explicit architectural reliance on epipolar projection or cost volumes, they still typically depend on pre-defined 3D representations such as tri-plane NeRFs, meshes, or 3DGS, along with their corresponding rendering equations. This reliance can limit their flexibility and overall potential.

An alternative line of work attempts to directly learn a geometry-free rendering function [70, 63, 68, 60, 37]. However, these approaches often face limitations in model capacity and scalability, which can hinder their ability to capture high-frequency details. Notably, Scene Representation Transformers (SRT) [63] aimed to avoid explicit, handcrafted 3D representations by learning a latent

scene representation via a transformer, an objective shared by our encoder-decoder architecture. Despite this similarity, certain design choices in SRT, such as its CNN-based token extractor and the use of cross-attention in the decoder, have been shown to lead to less effective performance. To address the issue, LVSM [31] proposes a method that is fully transformer-based, leveraging bidirectional self-attention for enhanced representational power. Furthermore, they introduce a novel and more scalable decoder-only architecture that directly learns the NVS function with minimal 3D inductive bias and without relying on an intermediate latent representation.

Our proposed method adopts the versatile and scalable decoder-only transformer backbone from LVSM, which has demonstrated its efficacy in NVS tasks by leveraging a data-driven approach with minimal handcrafted 3D inductive bias. However, a crucial distinction lies in the nature of our approach: unlike the deterministic LVSM, our method is generative. We aim to address the inherent limitations of deterministic methods by harnessing the generative capabilities of masked autoregressive diffusion models in an efficient manner.

## B.2 Generative-based NVS methods

The pursuit of generative-based (NVS) has recently seen significant advancements through the integration of diffusion models, drawing inspiration from successes in broader NVS [68, 63] and generative image-to-image tasks [61, 56, 62].

An early exploration in this domain was 3DiM [86], which trained image-to-image diffusion models for object-level multi-view rendering without explicit 3D representations. However, by training from scratch on limited 3D data, 3DiM's applicability was restricted to category-specific scenarios and lacked zero-shot generalization capabilities. Building on this, Zero-1-to-3 [43] adopted a similar geometry-free pipeline but significantly improved generalization and output quality by fine-tuning a pretrained 2D diffusion model on a larger 3D object dataset [15]. Despite these improvements, a key challenge for Zero-1-to-3 and other early image-based diffusion models for NVS (e.g., for distant viewpoints [65]) was multi-view inconsistency, as they typically generated each target view independently and probabilistically, leading to jitter or inconsistencies when rendering a camera trajectory.

To address this multi-view inconsistency, subsequent research diverged into several directions. One line of work focused on integrating explicit 3D inductive biases—such as 3D representations or epipolar attention—into the diffusion denoising process. Examples include SyncDreamer [44], ConsistNet [95], Consistent-1-to-N [97], and MegaScenes [75], though these often came at the cost of increased computation. Another set of approaches, including Instant3D [38], MVDream [66], and Wonder3D [46], aimed to predict a single grid of multiple, specific views simultaneously. While this improved consistency across those fixed views, it sacrificed the ability for fine-grained camera control. Works like MVDream [66], SyncDreamer [44], and more recently HexGen3D [48], generate multiple fixed views from a conditional image but do not support arbitrary viewpoint selection. To achieve consistent 3D object geometry from these image-based models, further steps like NeRF distillation, using techniques such as Score Distillation Sampling (SDS) [55, 64] or direct optimization on sampled images [89, 21], are often necessary. However, distillation techniques such as SDS can introduce substantial computational overhead due to test-time optimization.

More recently, a promising trend has emerged with models that jointly predict multiple target views while maintaining accurate camera control and ensuring view consistency, often through mechanisms like cross-view attention. This category includes methods such as Free3D [101], EscherNet [36], CAT3D [21], and SV3D [79]. Several video model-based approaches [85, 23, 99, 94, 102] also fall into this paradigm, increasing NVS performance. Despite these advancements, achieving high-quality generation with these recent models often necessitates substantial computational resources and extensive training data. Furthermore, their reliance on full-image iterative sampling typically results in slow inference times, limiting practical applicability. Our proposed method, UMAMI, addresses this critical issue by enabling photorealistic novel view rendering while maintaining efficient inference times.

Table 7: Hyperparameters for training `UMAMI`. We use the same set of hyperparameters for both RealEstate10K and DL3DV experiments.

| Component | Parameter | Value |
|---|---|---|
| Image Tokenizer | Image size | 256 |
| | Patch size | 8 |
| | Channels | 9 (3 RGB + 6 for Plücker) |
| Transformer | Layers | 24 |
| | Hidden dim | 768 |
| | Head dim | 64 |
| | QK Norm | True |
| Training | Batch size / GPU | 4 |
| | Num GPUS | 8 |
| | Learning rate | 0.0002 |
| | Optimizer $(\beta_1, \beta_2)$ | (0.9, 0.95) |
| | Grad clip norm | 3.0 |
| | Mixed precision | True |
| | Weight decay | 0.02 |
| | Train steps | 100k |
| | Warmup steps | 1000 |
| Data Setup | Input / Target views | 1 to 2 / 1 to 3 |
| | Center Crop | True |
| Loss Weights | L2 loss | 1.0 |
| | LPIPS loss | 0.0 |
| | Perceptual loss | 0.5 |
| | Diffusion loss | 10 |
| | Confidence loss | 1 |

## C  Implementation details

### C.1  Hyperparamters

We report the hyperparameters used in Table 7.

### C.2  Algorithm

We describe the sampling process of `UMAMI` in Algorithm 1.

---

**Algorithm 1** Hybrid Inference in UMAMI

---

**Require:** Trained model, context views $\{(I_{\text{ctx}}, \pi_{\text{ctx}})\}$, target pose $\pi_{\text{tgt}}$, threshold $\tau$, max unmasking steps $T_{\max}$

1: Tokenize context views into $\mathbf{c}$, initialize target tokens $\mathbf{x}$ with masked tokens
2: Encode $(\mathbf{c}, \mathbf{x})$ with Transformer to obtain latent $\mathbf{z}$
3: Predict confidence map $\mathbf{s}_p$ and patch-level scores $\mathbf{s}$
4: Partition target tokens:
- Deterministic tokens: $\mathbf{x}_D \leftarrow \{x_i \mid s_i \geq \tau\}$
- Stochastic tokens: $\mathbf{x}_S \leftarrow \{x_i \mid s_i < \tau\}$

5: Predict $\mathbf{x}_D$ in one pass using deterministic head: $\hat{\mathbf{x}}_D = \phi(\mathbf{z}_D)$
6: Compute sampling steps: $T_S = \lceil |x_S|/|x| \cdot T_{\max} \rceil$
7: **for** $t = T_S$ **to** $1$ **do**
8:     Sample random unmasked set $\mathbf{x}_t \subset \mathbf{x}_S$ following a cosine scheduler.
9:     Update $\mathbf{x}_t$ by DDPM sampling using $\varphi$ head.
10: **end for**
11: Merge $\hat{\mathbf{x}}_D$ and $\hat{\mathbf{x}}_S$ into full target image $\hat{I}_{\text{tgt}}$

---

# D  Additional quantitative results

## D.1  Multiple images generation

Table 8: Multi-view generation results on RealEstate10K.

| Dataset | # gen views | PSNR ↑ | LPIPS ↓ | SSIM ↑ |
|---|---|---|---|---|
| Re10K-2views-Extra | 1 | 28.95 | 0.107 | 0.897 |
|  | 3 | 28.65 | 0.109 | 0.892 |
| Re10K-2views-Interp | 1 | 28.85 | 0.101 | 0.899 |
|  | 3 | 28.52 | 0.105 | 0.894 |

As shown in Table 7, our model is trained to predict up to three target views simultaneously. This joint prediction encourages consistency across generated images. In Table 8, we report results for generating one and three views. The generation quality is comparable across both settings. Notably, we use a fixed number of unmasking steps ($T_{\max} = 32$) for all cases, which means generating multiple views in parallel can improve inference efficiency without sacrificing quality.

# E  Additional qualitative results

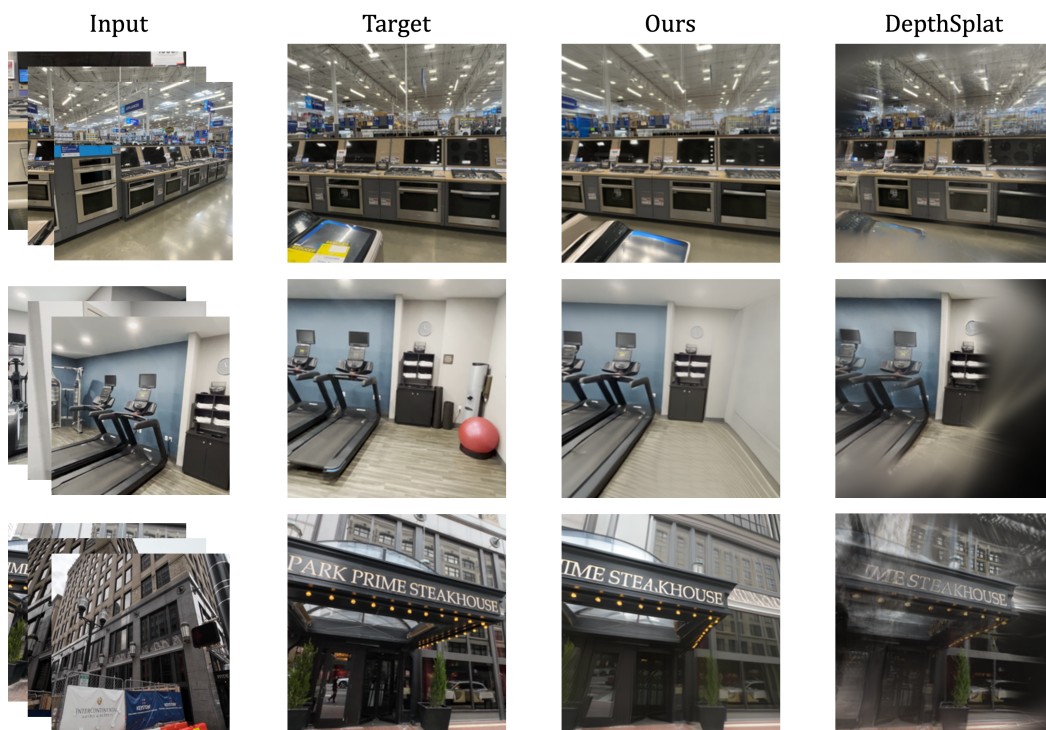

Figure 6: Additional qualitative comparisons on DL3DV dataset.

Additional qualitative evaluations are presented on the DL3DV dataset [41], where our method is compared against DepthSplat [91] under a three-view input configuration (Figure 6). As depicted UMAMI demonstrates notably sharp rendering, particularly in unobserved regions. This is achieved by leveraging its generative capabilities to synthesize plausible details unobserved region of input images.

Furthermore, to investigate the impact of the diffusion threshold hyperparameter, $\tau$, on UMAMI's performance, its value was systematically varied, with findings illustrated in Figure 7. An initial setting of $\tau = 0$, corresponding to a fully deterministic operation of UMAMI, achieved rapid inference.

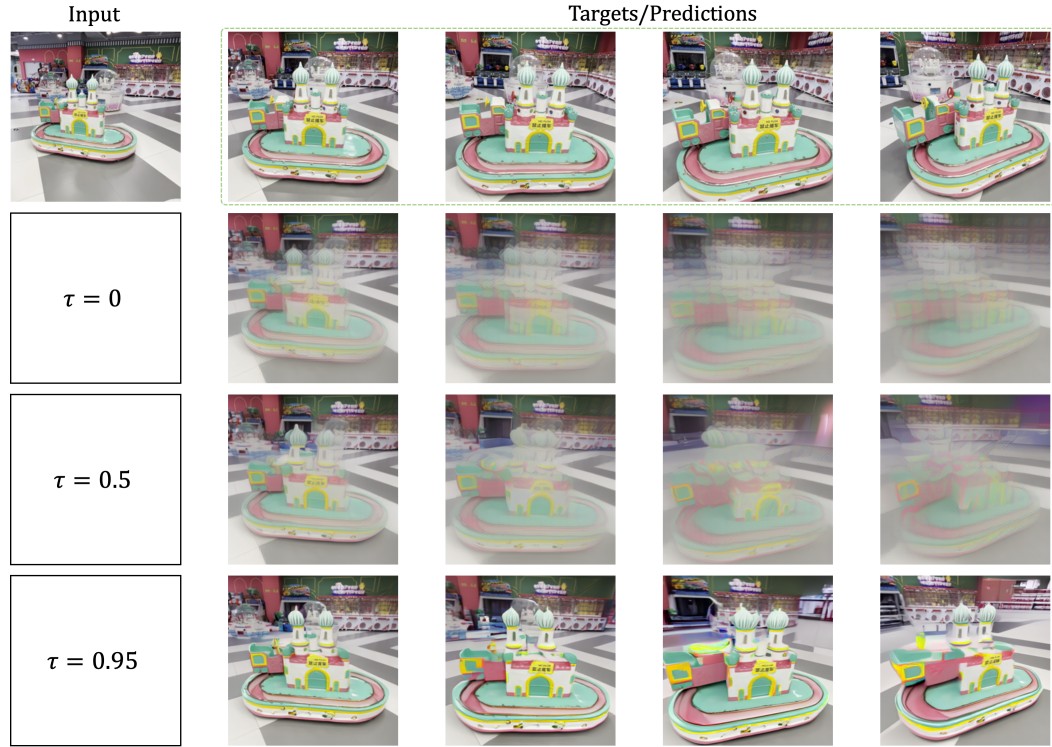

Figure 7: Impact of the diffusion threshold hyperparameter $\tau$ on rendering outcomes. The top row shows the single input view alongside four corresponding target views. The subsequent rows (2-4) illustrate the results as $\tau$ is incrementally increased. While a lower $\tau$ promotes deterministic behavior and faster inference, higher values of $\tau$ lead to notably sharper image rendering quality.

However, this configuration resulted in image blurring, an artifact attributable to unobserved regions in the input view. Progressively increasing $\tau$ to $0.5$ and subsequently to $0.95$ yielded a significant enhancement in rendering quality. This improvement, however, was accompanied by an increase in running time. Finally, to demonstrate the complete sampling dynamics of our method, the unmasking processes for $\tau = 0.95$ and for full unmasking diffusion process ($\tau = 1$) are presented in the supplementary video.

