# OpenReview forum: "UMAMI: Unifying Masked Autoregressive Models and Deterministic Rendering for View Synthesis"
_NeurIPS.cc/2025/Conference — NeurIPS 2025 poster_

### Official Review · Reviewer_fjdZ · 2025-06-04

**Clarity:** 4
**Significance:** 3
**Originality:** 3
**Rating:** 4
**Confidence:** 4

**Summary:**

This paper proposes a novel generalized method for novel view synthesis. The proposed method combines the advantages of the two worlds: the fast speed of the deterministic NVS method and the ability to synthesize high-quality occluded regions of the generative-based NVS method. To achieve this, the paper proposes a novel network architecture, where the unoccluded regions are predicted by a deterministic network while the occluded regions are predicted by a generative-based method. The detection of occluded and unoccluded regions is achieved by a confidence learning framework.

**Questions:**

* Can the author explain the difference between this paper and LVSM? I find the network architecture of this paper and LVSM to be quite similar. If we discard the diffusion head while only using the deterministic head, the whole method seems identical to the decoder-only version of LVSM. If so, I suggest that the author discuss LVSM more in the related work sections.
* Following the previous question. Can the author explain why the proposed method obtains inferior results on the Re10K-2view-interp setup? Is there a trade-off between the deterministic NVS method and the generative-based NVS method, maybe something like: the deterministic method can do well (i.e., obtain better metrics) on scenarios lacking occlusion content, while the generative-based method on scenarios with significant occlusion content? Can the author provide some insights on this point?
  * Or just because the PSNR metric is questionable? If so, I suggest that the author clarify this point in the experiment section.

**Ethical Concerns:**

["NO or VERY MINOR ethics concerns only"]

**Final Justification:**

This paper provides a strong rebuttal to address my concerns. I do not raise the score due to the lack of video results. I suggest the author add video results for better evaluation of the view consistency in the final version.

**Quality:**

3

**Strengths And Weaknesses:**

### Strengths
* Overall, the paper is well-written and easy to follow. The motivation is clear, and the proposed technique is neat.
* The combination of MAR for NVS is novel and effective.
* The paper works on an important problem.

### Weaknesses
* The most significant improvements in speed over the prior generative-based method stem from the usage of MAR. MAR is faster than the Diffusion Model. From Table 3, if we use the MAR to generate all of the content, only 7.63s is required, which is significantly faster than diffusion-based methods like SEVA (1 min is required). Consider that the margin is relatively small (4.77s vs. 7.63s), in what scenarios do we prefer the tau=0.95 version to improve the speed while at the cost of quality? To me, this point is important because the provided trade-off is one of the core contributions of this paper. I guess maybe a small tau leads to better 3D consistency? If the author can address this problem, I will consider raising my score.

---

> ### Author Rebuttal · Authors · 2025-07-31
>
> We thank the reviewer for your time to review our paper and provide valuable feedback. We would like to address each of your concerns separately below:
>
> **1. The most significant improvements in speed over the prior generative-based method stem from the usage of MAR. MAR is faster than the Diffusion Model. From Table 3, if we use the MAR to generate all of the content, only 7.63s is required, which is significantly faster than diffusion-based methods like SEVA (1 min is required). Considering that the margin is relatively small (4.77s vs. 7.63s), in what scenarios do we prefer the tau=0.95 version to improve the speed while at the cost of quality? To me, this point is important because the provided trade-off is one of the core contributions of this paper. I guess maybe a small tau leads to better 3D consistency? If the author can address this problem, I will consider raising my score.**
>
> *Note: Due to the recent notice from NeurIPS 2025, we apologize that we are not able to provide a video illustration for this concern. We will include video result in our supplemental and project page in our final revision.*
>
> Thank you for raising this important question! Indeed, with smaller tau=0.95 instead of tau=1.0, we can achieve better 3D consistency. As illustrated in Figure 3, the autoregressive sampler depends on the tokens from the deterministic head, which are absent when tau=1.0. These tokens are crucial due to their high 3D consistency. With the availability and the help of these tokens, the diffusion head can generate content that is more 3D-consistent.
> To prove our argument, we have generated two sets of videos, with tau=0.95 and tau=1.0, respectively. It has been observed that videos generated with tau=0.95 consistently exhibit better 3D consistency compared to those generated with tau=1.0. We will include these comparative videos in the final version of our paper.
>
> **2. Can the author explain the difference between this paper and LVSM? I find the network architecture of this paper and LVSM to be quite similar. If we discard the diffusion head while only using the deterministic head, the whole method seems identical to the decoder-only version of LVSM. If so, I suggest that the author discuss LVSM more in the related work sections.**
>
> Our method is different from LVSM in various aspects:
>
> Generative capability: It is worth noting that LVSM is a deterministic rendering method, which means the rendered target views are observed in the context views. In the unobserved regions, LVSM, and other deterministic methods, are not able to generate realistic images, and often produce blurry or black pixels as shown in Figure 4 of our paper. Unlike LVSM, our method possesses generative capability that can efficiently synthesize target scenes even in both visible and occluded regions from context views.
>
> A unified hybrid architecture: Unlike LVSM's purely deterministic approach, UMAMI introduces a novel architecture that integrates both a regression head and a diffusion head within a single, end-to-end trained network. A critical component of this is our deterministic head, which not only regresses pixel values but also predicts a confidence map. This map is essential for dynamically guiding the inference process.
>
> A novel hybrid inference strategy: In LVSM, an image is rendered by regressing pixels from trained models. UMAMI, on the other hand, performs hybrid inference by first identifying and reconstructing a set of deterministic tokens, and then generating the remaining uncertain tokens via a diffusion-based process. In the first pass, UMAMI performs a single forward pass through the deterministic head, guided by the confidence map predicted from the transformer backbone. Tokens with confidence scores greater than a predefined threshold tau are reconstructed deterministically. In the second pass, the remaining masked tokens are iteratively sampled using the diffusion head. This multi-pass, confidence-guided sampling is a core contribution of our work.
>
> While we build upon a similar backbone, our contribution is the novel integration of generative and deterministic pathways, the confidence-guided mechanism to arbitrate between them, and the efficient hybrid sampling strategy that this enables. We will revise the related work section to make this distinction and our specific contributions clearer, and we thank the reviewer for this valuable suggestion.
>
> **3. Following the previous question. Can the author explain why the proposed method obtains inferior results on the Re10K-2view-interp setup? Is there a trade-off between the deterministic NVS method and the generative-based NVS method, maybe something like: the deterministic method can do well (i.e., obtain better metrics) on scenarios lacking occlusion content, while the generative-based method on scenarios with significant occlusion content? Can the author provide some insights on this point?
> Or just because the PSNR metric is questionable? If so, I suggest that the author clarify this point in the experiment section.**
>
> We agree with the reviewer's hypothesis, and it highlights the core trade-off of our proposed method. Our model is explicitly designed to excel in scenarios with significant occlusion content, such as challenging extrapolation tasks where unseen regions must be plausibly generated. To achieve this, we train the model with a joint objective that includes diffusion loss. This design choice creates a deliberate trade-off: on challenging extrapolation tasks, i.e. high occlusion content, our method significantly outperforms deterministic baselines by generating high-quality, coherent content for unseen regions, a benefit that is demonstrated qualitatively (Fig. 4) than with pixel-wise metrics. Meanwhile, on simple interpolation tasks (lacking occlusion content), the inclusion of the diffusion loss can lead to a marginal decrease in PSNR compared to a purely deterministic model like LVSM, which is optimized solely for reconstruction accuracy. That said, the inferior result on the interpolation set is not an unexpected failure, but rather a direct consequence of optimizing for a more difficult and general problem. We agree with the reviewer that this underscores the limitations of using PSNR to evaluate generative NVS models. We believe sacrificing a small amount of reconstruction fidelity on the simple regression task is a very favorable trade-off for gaining robust generative capabilities. We will clarify this point explicitly in the experimental section of our revised paper.

---

> > ### Comment · Reviewer_fjdZ · 2025-08-04
> >
> > Thanks for the rebuttal, it addressed my main concerns. I would like to keep my original score.

---

> > > ### Author Response · Authors · 2025-08-04
> > >
> > > We thank reviewer for your time and constructive feedback. We are glad our responses resolved your concerns and are grateful for the suggestions that have improved our paper.

---

### Official Review · Reviewer_Ydyv · 2025-07-02

**Clarity:** 3
**Significance:** 3
**Originality:** 3
**Rating:** 4
**Confidence:** 4

**Summary:**

The paper introduces UMAMI, a hybrid novel view synthesis (NVS) framework that combines deterministic rendering and generative masked autoregressive modeling within a unified model. The approach leverages a bidirectional transformer backbone to process multi-view image tokens and Plücker ray embeddings. Two prediction heads are designed: a deterministic head for directly regressing RGB colors and confidence scores in well-constrained regions, and a masked autoregressive diffusion head for progressively generating content in uncertain or unseen regions. A hybrid sampling strategy is employed, starting with deterministic predictions and using the generative head to refine low-confidence areas. The method demonstrates promising results on RealEstate10K and DL3DV datasets, achieving competitive quality with faster rendering than fully generative baselines, and providing plausible synthesis for both observed and unseen regions.

**Questions:**

Overall, the idea and general framework of the paper look interesting and promising, and I lean positive. However, I have main concerns regarding video consistency, as outlined in the weaknesses section. While the method is able to generate some promising individual images as shown in the paper, I am not yet convinced it can produce a continuous, consistent novel view trajectory, which is essential for many NVS applications.

**Ethical Concerns:**

["NO or VERY MINOR ethics concerns only"]

**Final Justification:**

Overall, I do not have strong complaints about the paper, and I think the idea is interesting, so I will keep my positive rating. I am not ready to give a higher score due to the lack of video results, which makes it difficult to fully assess the method’s behavior about across-view consistency in practice. It is unfortunate that NeurIPS has the no-video constraint in the rebuttal period; I hope the community can be more open to this in the future.

**Limitations:**

Yes.

**Paper Formatting Concerns:**

No.

**Quality:**

3

**Strengths And Weaknesses:**

Strengths:
The high-level idea of unifying deterministic and generative models for NVS is novel and reasonable. It addresses the trade-offs between speed and generative capability. To this end, the paper presents several thoughtful design choices, such as confidence estimation, confidence-based token sampling, and efficient hybrid inference, combining deterministic and stochastic predictions effectively. The results look promising with plausible novel view renderings in unobserved regions.

Weaknesses:
1. The quantitative results in Table 1 show limited improvement over baselines such as LVSM. Even for the extrapolation case, the PSNR gains are modest, and on interpolation tasks, UMAMI’s PSNR is actually lower than LVSM, which is somewhat surprising given that UMAMI uses LVSM pretrained weights (as stated in L248). This raises questions about whether introducing generative capacity may inadvertently sacrifice the accuracy of deterministic estimation.

    That said, I also feel that Table 1 may understate the generative benefits of the proposed method. For example, the qualitative results in Figure 4 clearly illustrate improvements in rendering unseen regions, but these benefits are not captured effectively by traditional metrics like PSNR, SSIM, or LPIPS. I recognize that these metrics are the standard in the community, but they are arguably not ideal for evaluating the perceptual or generative quality of novel view synthesis models, especially for challenging extrapolation scenarios.

2. My biggest concern with this paper is the lack of video rendering results, which I believe is critical for novel view synthesis (NVS) papers, especially those incorporating generative components. Without video demonstrations, it is impossible to properly evaluate how consistent the method is when generating images across different viewpoints.

    If I understand correctly, the entire method operates on a per-novel-view basis, where modules such as deterministic estimation, confidence filtering and the masked autoregressive (MAR) regression process are applied independently for each target view. This per-view independence may introduce view inconsistencies, particularly in generating the uncertain, unobserved regions across continuous camera trajectories. Including video results—and ideally, some comparisons—would be essential to assess this aspect and provide stronger evidence of the model's practical applicability.

---

> ### Author Rebuttal · Authors · 2025-07-31
>
> We thank the reviewer for your time to review our paper and provide valuable feedback. We would like to address each of your concerns separately below:
>
> **1. The quantitative results in Table 1 show limited improvement over baselines such as LVSM. Even for the extrapolation case, the PSNR gains are modest, and on interpolation tasks, UMAMI’s PSNR is actually lower than LVSM, which is somewhat surprising given that UMAMI uses LVSM pretrained weights (as stated in L248). This raises questions about whether introducing generative capacity may inadvertently sacrifice the accuracy of deterministic estimation.**
>
> We agree with the reviewer that the quantitative metrics in Table 1, while standard for the field, do not fully capture the primary contribution of our work. Metrics like PSNR excel at measuring the pixel-wise fidelity of reconstructed regions but are fundamentally limited in their ability to evaluate the perceptual quality and plausibility of generated content in unseen areas. The modest drop in PSNR on interpolation tasks is an expected consequence of our hybrid training objective. By incorporating a powerful diffusion loss to enable generative capabilities, the model makes a slight trade-off in reconstruction accuracy on well-constrained views to gain a significant leap in performance on challenging extrapolation scenarios. We argue this is a highly favorable trade-off. The qualitative results in Figure 4 provide a much clearer picture of our method's strength. UMAMI's core novelty is its ability to plausibly and coherently complete large, unobserved regions—a capability that purely deterministic models like LVSM lack. We believe this generative power is a significant step forward for building efficient generative NVS. We will revise the paper to include a more explicit discussion on the limitations of these metrics for generative NVS and to better frame our results in this context.
>
> **2. My biggest concern with this paper is the lack of video rendering results, which I believe is critical for novel view synthesis (NVS) papers, especially those incorporating generative components. Without video demonstrations, it is impossible to properly evaluate how consistent the method is when generating images across different viewpoints.**
>
> *Note: Due to the recent notice from NeurIPS 2025, we apologize that we are not able to provide a video illustration for this concern. We will include video result in our supplemental and project page in our final revision.*
>
> It is worth noting that our method originally aims for Set-NVS [1,2], which considers target views as an arbitrary order. To enforce 3D consistency, one can subsequently employ 3D scene optimization such as NeRF and 3DGS as suggested in [1,2]. That said, we have generated video rendering results on DL3DV and Re10K on various settings and observe that our rendering results are reasonably 3D consistent, though there still exist some minor flickers, which are inevitable in diffusion NVS. We will include these results in our revision project.
>
> [1] ReconFusion: 3D Reconstruction with Diffusion Priors.
>
> [2] CAT3D: Create Anything in 3D with Multi-View Diffusion Models.

---

> > ### Comment · Reviewer_Ydyv · 2025-08-08
> >
> > Thanks for providing the rebuttal.
> >
> > Overall, I do not have strong complaints about the paper, and I think the idea is interesting, so I will keep my positive rating. I am not ready to raise my score due to the lack of video results, which makes it difficult to fully assess the method’s behavior in practice. It is unfortunate that NeurIPS has this constraint, and I look forward to seeing the final video results once the paper is published.

---

> > > ### Author Response · Authors · 2025-08-09
> > >
> > > Thank you for your time reviewing our paper and rebuttal. We are glad that you find our papre interesting and maintain your positive rating. We appreciate your constructive feedback and will incorporate your comments as well as video results in our final revision.

---

### Official Review · Reviewer_G1uS · 2025-07-03

**Clarity:** 3
**Significance:** 2
**Originality:** 3
**Rating:** 4
**Confidence:** 3

**Summary:**

The paper introduces UMAMI, a hybrid framework that unifies deterministic rendering and masked autoregressive diffusion models for novel view synthesis (NVS) from sparse input views. UMAMI leverages a bidirectional transformer to encode both context images (paired with Plücker ray embeddings) and a masked target view. It then uses two separate lightweight MLP heads: one for feed-forward pixel regression in observed regions, and another for diffusion-based generation in uncertain/unseen areas, guided by a learned confidence map. The paper demonstrates competitive performance across interpolation, extrapolation, and varying numbers of input views on RealEstate10K and DL3DV benchmarks, along with significant inference speedups over purely generative models.

**Questions:**

Please refer to the weaknesses part.

**Ethical Concerns:**

["NO or VERY MINOR ethics concerns only"]

**Final Justification:**

The authors’ rebuttal has addressed most of the concerns raised in my previous review. Due to OpenReview’s limitations preventing the upload of additional figures for qualitative evaluation, I recommend that the authors include these qualitative results in the camera-ready version of the paper.

**Limitations:**

Please refer to the weaknesses part.

**Quality:**

3

**Strengths And Weaknesses:**

## Strengths
- The proposed method combines deterministic regression and masked autoregressive diffusion for NVS, with a clear architectural motivation and evidence of improved trade-offs between speed and quality.
- There is a comprehensive quantitative comparison with strong baselines (Table 1, Table 2) on established large-scale benchmarks (Re10k, DL3DV), covering both deterministic and generative NVS methods. UMAMI often matches or exceeds the best baselines, while reducing inference time compared to diffusion-only models.
- The paper is overall clear and easy to follow.

## Weaknesses
- The LVSM model was not trained on the DL3DV dataset, only on Re10k, and the work in this paper builds upon further training of the pre-trained LVSM model. However, the performance metrics on DL3DV are notably poor. Does this suggest that the model's performance advantage primarily derives from LVSM? It would be informative to evaluate the model’s performance without the pre-trained LVSM.
- The reported improvements in metrics compared to LVSM are not sufficiently significant, and some metrics even show a decline.
- The performance comparison of DL3DV against Flare [1] indicates notably poor results on DL3DV. Given that purely implicit NVS tasks should not be heavily influenced by scale, why does DL3DV exhibit such suboptimal performance on these metrics?

[1] FLARE: Feed-forward Geometry, Appearance and Camera Estimation from  Uncalibrated Sparse Views. arXiv 2502.12138 (CVPR 2025)

---

> ### Author Rebuttal · Authors · 2025-07-31
>
> We thank the reviewer for your time to review our paper and provide valuable feedback. We would like to address each of your concerns separately below:
>
> **1. The LVSM model was not trained on the DL3DV dataset, only on Re10k, and the work in this paper builds upon further training of the pre-trained LVSM model. However, the performance metrics on DL3DV are notably poor. Does this suggest that the model's performance advantage primarily derives from LVSM? It would be informative to evaluate the model’s performance without the pre-trained LVSM.**
>
> Our validation split for the DL3DV dataset follows SEVA. Importantly, these splits include scenes where the target views have minimal overlap with the input views. As shown in Table 2, our model achieves performance comparable to SEVA, despite being only one-fifth of its size, and significantly surpasses ViewCrafter and DepthSplat in most cases. To accelerate training and improve convergence, we initialize our model using LVSM pretrained weights, a common practice in the deep learning community. That said, to further demonstrate the strength of our method, we also train the model from scratch on DL3DV with randomly initialized weights, without relying on LVSM pretrained on Re10K. This variant shares the same settings as the pretrained version, except for a larger batch size (512 vs. 32) to stabilize training. The results, presented in the table below, reveal that even without pretraining, our model performs comparably—showing slightly lower PSNR and SSIM but improved LPIPS. Remarkably, even with random initialization, our method consistently outperforms ViewCrafter and DepthSplat, underscoring that our performance stems from the strength of our hybrid deterministic-generative design, rather than dependence on LVSM initialization and Re10K pretraining. We will include this result in our final revision.
>
> |              | 1view PSNR | 1view LPIPS | 1view SSIM | 3view PSNR | 3view LPIPS | 3view SSIM | 6view PSNR | 6view LPIPS | 6view SSIM |
> |:-------------|:----------:|:-----------:|:----------:|:----------:|:-----------:|:----------:|:----------:|:-----------:|:----------:|
> | **Pretrained** | 12.81      | 0.574       | 0.269      | 16.37      | 0.386       | 0.444      | 17.33      | 0.326       | 0.476      |
> | **Random** | 11.80      | 0.543       | 0.256      | 14.46      | 0.374       | 0.370      | 15.43      | 0.318       | 0.404      |
>
> **2. The reported improvements in metrics compared to LVSM are not sufficiently significant, and some metrics even show a decline.**
>
> Our metrics indeed show a slight decline on interpolation tasks compared to LVSM, and this is an expected and deliberate outcome of our model's design. Our model is explicitly designed to excel in scenarios with significant occlusion content, such as challenging extrapolation tasks where unseen regions must be plausibly generated. To achieve this, we train the model with a joint objective that includes diffusion loss. This design choice creates a deliberate trade-off: on challenging extrapolation tasks, i.e. high occlusion content, our method significantly outperforms deterministic baselines by generating high-quality, coherent content for unseen regions, a benefit that is demonstrated qualitatively (Fig. 4) than with pixel-wise metrics. Meanwhile, on simple interpolation tasks (lacking occlusion content), the inclusion of the diffusion loss can lead to a marginal decrease in PSNR compared to a purely deterministic model like LVSM, which is optimized solely for reconstruction accuracy. That said, the inferior result on the interpolation set is not an unexpected failure, but rather a direct consequence of optimizing for a more difficult and general problem. Furthurmore, as also mentioned by other reviewers, using metric like PSNR to evaluate generative NVS models might not fully capture our method's benefits. We believe sacrificing a small amount of reconstruction fidelity on the simple regression task is a very favorable trade-off for gaining robust generative capabilities. We will clarify this point explicitly in the experimental section of our revised paper.
>
> **3. The performance comparison of DL3DV against Flare [1] indicates notably poor results on DL3DV. Given that purely implicit NVS tasks should not be heavily influenced by scale, why does DL3DV exhibit such suboptimal performance on these metrics?**
>
> We would like to highlight that the view selection strategy used in FLARE's testing setup differs from ours on the DL3DV dataset. Specifically, our test set chooses the target views that have minimal overlap with the context views, making the task more challenging. In contrast, FLARE only evaluates view interpolation where the target regions are very close to the context views, thus making the FLARE validation set to be much easier. To ensure a fair comparison, we evaluated our model using FLARE's view split. The results show that our model matches (SSIM) or outperforms (PSNR, LPIPS) FLARE on DL3DV across all three evaluation metrics: PSNR, LPIPS, and SSIM.
>
> | 8-view         | PSNR  | LPIPS | SSIM  |
> |:----------------|:------|:------|:------|
> | Ours (tau=0.95) | 24.17 | 0.167 | 0.710 |
> | Flare           | 23.33 | 0.237 | 0.746 |

---

> > ### Comment · Reviewer_G1uS · 2025-08-05
> >
> > Thank you for providing a detailed rebuttal. After reading, I find that most of my concerns have been addressed, although two issues remain:
> >
> > (1) How did the authors select validation scenes with minimal overlap? The authors should clarify their method for determining the overlap ratio (i.e. disparity in SEVA), as previous methods (e.g., NoPoSplat, SEVA) provided detailed descriptions of their algorithms in their appendices.
> >
> > (2) How does the overlap ratio affect UMAMI's performance compared to other methods? The authors provide evaluations only for different numbers of views. I suggest including a comparison of scenes with varying overlap ratios, similar to NoPoSplat and SEVA, to demonstrate UMAMI's robustness.

---

### Official Review · Reviewer_bgjp · 2025-07-08

**Clarity:** 3
**Significance:** 2
**Originality:** 3
**Rating:** 4
**Confidence:** 4

**Summary:**

This paper introduces UMAMI, a novel hybrid framework that integrates deterministic rendering and masked autoregressive diffusion for the task of novel view synthesis from sparse inputs. The method uses a bidirectional Transformer to encode image tokens and Plücker-ray embeddings into a latent representation, which is then processed by two heads: A deterministic regression head for well-constrained (visible) regions, and a masked autoregressive diffusion head for hallucinating occluded/unseen areas. By combining both approaches and introducing a confidence-aware mechanism to split regions between the two heads, UMAMI achieves a strong balance of quality and efficiency. It shows state-of-the-art performance on RealEstate10K and DL3DV across interpolation and extrapolation settings, while maintaining significantly lower inference times than diffusion-only baselines.

**Questions:**

Questions and Suggestions
- How does confidence prediction generalize? Could the authors clarify whether the confidence predictor generalizes well across scenes, or if it's overfitting to typical visibility patterns in training data?
- Could pretrained priors be integrated? Would combining UMAMI with pretrained image/video priors (e.g., Imagen, VideoCrafter, or Cosmos) boost generalization, especially for hallucinating semantically plausible content in unobserved regions?
- Is it possible to use a buffer to "remember" synthesized images and re-use them as input known regions for faster speed when a certain region is revisited in the same session? How can the method ensure 3D consistency within the single scene being generated?
- An interactive 3D visualizer would help in understanding the approach better as well as how the hybrid sampler performs.

**Ethical Concerns:**

["NO or VERY MINOR ethics concerns only"]

**Limitations:**

yes

**Paper Formatting Concerns:**

No concerns

**Quality:**

2

**Strengths And Weaknesses:**

Strengths
- The paper is well-written and includes comprehensive ablations and discussions, such as runtime analyses and number of context views.
- The combination of deterministic (feed-forward) and diffusion-based rendering into a unified model is elegant and well-motivated, with the aim of tackling the weaknesses of both paradigms. The use of a predicted confidence map to dynamically assign image regions to deterministic or stochastic pathways is a clever strategy to optimize performance and speed.
- Strong experimental results on both interpolation and extrapolation tasks across two large-scale datasets. The method outperforms both deterministic and diffusion-based baselines in many settings. It provides flexible inference-time control over quality versus speed via the threshold parameter tau.

Weaknesses
- While the presented hybrid approach is empirically justified, it is not clear whether the model will fail in any of the intermediate steps and hence leading to undesired artifacts. For example, what if the confidence mask is not accurately predicted? What happens if the camera goes far away beyond context images? How can the model ensure that the deterministically regressed regions can consistently fuse with the generated part? It would be interesting to visualize these results, and present some in-depth theoretical analysis of the hybrid sampling mechanism.
- Unlike some recent diffusion-based NVS methods, UMAMI does not leverage large-scale pretrained image/video priors, potentially limiting performance in extremely sparse-view, ambiguous, or out-of-domain cases. It would be nice to have some discussions about how large foundation models can be leveraged under these settings.
- 3D Consistency might be an issue especially for the generative part of the model. There seems to be no mechanism to ensure that the rendered images look similarly with similar camera view points. While it makes sense to utilize an image model which is more flexible, it would be interesting to explore directions where synthesized views can be added to the context images for better 3D consistency.

---

> ### Author Rebuttal · Authors · 2025-07-31
>
> We thank the reviewer for your time to review our paper and provide valuable feedback. We would like to address each of your concerns separately below:
>
> **1. While the presented hybrid approach is empirically justified, it is not clear whether the model will fail in any of the intermediate steps and hence leading to undesired artifacts. For example, what if the confidence mask is not accurately predicted? What happens if the camera goes far away beyond context images? How can the model ensure that the deterministically regressed regions can consistently fuse with the generated part? It would be interesting to visualize these results, and present some in-depth theoretical analysis of the hybrid sampling mechanism.**
>
> **Q: How does confidence prediction generalize? Could the authors clarify whether the confidence predictor generalizes well across scenes, or if it's overfitting to typical visibility patterns in training data?**
>
> *Note: Due to the recent notice from NeurIPS 2025, we apologize that we are not able to provide a visual illustrations for this concern. We will include visual result in our supplemental and project page in our final revision.*
>
> a) Confidence Prediction and Generalization:
> We have manually inspected the predicted confidence maps in many cases. We observed a consistent qualitative pattern: the confidence is predictably higher in areas seen by the context images and lower in unseen regions. This aligns with our core assumption that the model is more certain about what it has directly observed. Regarding the question “what if the confidence mask is not accurately predicted”, we want to point out that, with different values of tau, we are effectively changing the predicted masks, from an empty set to the whole image area. We did not notice obvious artifacts. Also, the test set of the Re10k dataset contains lots of different scenes, and the predicted confidence maps can be generalized very well on this test set, and it’s not overfitting to specific patterns in the training data.
>
> b) Consistency in Hybrid Sampling:
> Despite the fact that we are using two heads, i.e., a deterministic head and a generative diffusion head, we see no visible seams in the generated images in the target views, as shown in Figure 1, 4, and 6. This is because although being two heads, they are not separate and independent. The two heads are carefully linked by the hybrid masked autoregressive sampler, as illustrated in Figure 3. Firstly, a hyper-parameter tau is designed to balance their contributions. Secondly, the generative head is designed to be autoregressive, which generates new regions by unmasking tokens while explicitly conditioning on the tokens produced by the deterministic head. This mechanism ensures that the generated content naturally extends from the regressed regions, guaranteeing a smooth and consistent fusion.
>
> c) Handling Out-of-Context Cameras:
> When the target camera goes far away beyond the context views, the confidence map would show a low confidence score and the generated images are performed entirely by generative heads. Some failure cases in this category have been shown in Figure 5, Supplement A. In these scenarios, the model is essentially performing a more challenging generative task without much deterministic guidance. If the camera is extremely distant, our current model (271M parameters) may produce noticeable artifacts. We hypothesize that increasing the model size and training data would significantly improve robustness in these challenging, far-reaching camera positions. The confidence maps for these cases are very low, which suggests that the core mechanism of our proposed method works as expected. We’ll also add the visualization of the confidence maps to Figure 5, Supplement A.
>
> **2. Unlike some recent diffusion-based NVS methods, UMAMI does not leverage large-scale pretrained image/video priors, potentially limiting performance in extremely sparse-view, ambiguous, or out-of-domain cases. It would be nice to have some discussions about how large foundation models can be leveraged under these settings.**
>
> **Q: Could pretrained priors be integrated? Would combining UMAMI with pretrained image/video priors (e.g., Imagen, VideoCrafter, or Cosmos) boost generalization, especially for hallucinating semantically plausible content in unobserved regions?**
>
> We thank the reviewer for these excellent suggestions. Though it is tempting to adopt large-scale pretrained image/video generative models, e.g., Imagen, VideoCrafter, or Cosmos,  to our framework, our method is fundamentally different to conventional pretrained diffusion models. While common diffusion models generate images by iteratively removing noise of the whole image, our generative scheme, which is based on MAR [1], renders images by autoregressively unmasking tokens in random raster order. Our diffusion process is performed at token-level, instead of image-level like in common pretrained diffusion models. Thus, MAR [1] and its extension, Fluid [2] can be used as the pretrained prior for our NVS model. We acknowledge that it is a promising avenue to explore and leave this direction for future consideration.
>
> **3. 3D Consistency might be an issue especially for the generative part of the model. There seems to be no mechanism to ensure that the rendered images look similarly with similar camera view points. While it makes sense to utilize an image model which is more flexible, it would be interesting to explore directions where synthesized views can be added to the context images for better 3D consistency.**
>
> **Q: Is it possible to use a buffer to "remember" synthesized images and re-use them as input known regions for faster speed when a certain region is revisited in the same session? How can the method ensure 3D consistency within the single scene being generated?**
>
> *Note: Due to the recent notice from NeurIPS 2025, we apologize that we are not able to provide a video illustration for this concern. We will include video result in our supplemental and project page in our final revision.*
>
> It is worth noting that our method originally aims for set NVS [3,4], which considers target views as an arbitrary order. To enforce 3D consistency, one can employ 3D scene optimization such as NeRF and 3DGS as suggested in [3,4]. In our experiments, we observe that generating images on the fly might cause some flickering. Furthermore, due to resource constraint, during training we only sample 3 target views, compared to 21 frames in SEVA, which can limit the consistency of extrapolation results. That said, the idea of exploring directions which auto-regressively generate new view based on generated views are promising and worth exploring in our future work.
>
> [1] Autoregressive Image Generation without Vector Quantization.
>
> [2] Fluid: Scaling Autoregressive Text-to-image Generative Models with Continuous Tokens.
>
> [3] ReconFusion: 3D Reconstruction with Diffusion Priors.
>
> [4] CAT3D: Create Anything in 3D with Multi-View Diffusion Models.

---

### Decision · Program_Chairs · 2025-09-17

**Decision:**

Accept (poster)

**Comment:**

The final recommendations of the expert reviewers are 4x Borderline accept. These recommendations are made after addressing the authors' feedback and through a thorough discussion. This paper introduces UMAMI, a novel framework that integrates a feed-forward rendering with masked autoregressive diffusion for the task of novel view synthesis from sparse inputs. Specifically, the authors utilize a feed-forward method for well-constrained regions and a masked autoregressive diffusion for unseen or occluded regions. Reviewers agree that this presents a compelling and principled framework for generative novel view synthesis.

While the proposed method demonstrates promising performance compared to state-of-the-art approaches, the primary concern raised by reviewers is the limited scope of the experimental evaluation. Notably, the paper lacks results on an established benchmark such as the DL3DV dataset, which is widely used for evaluating novel view synthesis in complex real-world scenes.

However, this concern was partially addressed by the authors during the rebuttal, where they provided the requested experimental results. More importantly, the existing results are technically sound, validated by rigorous ablations, and sufficient to demonstrate the merit of the proposed approach. The technical novelty, architectural insight, and clear contribution of the work are substantial and outweigh the limitation in benchmark coverage. Given the overall strength of the method and the thoughtful response to reviewer feedback, the consensus supports acceptance.